# AAV-ie enables safe and efficient gene transfer to inner ear cells

Fangzhi Tan [1,14], Cenfeng Chu [1,2,3,14], Jieyu Qi [4,5,14], Wenyan Li [6,14], Dan You [6], Ke Li [3,7,8], Xin Chen [2,3], Weidong Zhao [6], Cheng Cheng [9,10], Xiaoyi Liu [1,2,3], Yunbo Qiao [11], Bing Su [1,2,3], Shuijin He [2], Chao Zhong [7], Huawei Li [6,15], Renjie Chai [4,5,6,12,13,15] & Guisheng Zhong [1,2,5,15]

Hearing loss is the most common sensory disorder. While gene therapy has emerged as a promising treatment of inherited diseases like hearing loss, it is dependent on the identification of gene delivery vectors. Adeno-associated virus (AAV) vector-mediated gene therapy has been approved in the US for treating a rare inherited eye disease but no safe and efficient vectors have been identified that can target the diverse types of inner ear cells. Here, we identify an AAV variant, AAV-inner ear (AAV-ie), for gene delivery in mouse inner ear. Our results show that AAV-ie transduces the cochlear supporting cells (SCs) with high efficiency, representing a vast improvement over conventional AAV serotypes. Furthermore, after AAV-ie-mediated transfer of the *Atoh1* gene, we find that many SCs trans-differentiated into new HCs. Our results suggest that AAV-ie is a useful tool for the cochlear gene therapy and for investigating the mechanism of HC regeneration.

[1] iHuman Institute, ShanghaiTech University, 201210 Shanghai, China. [2] School of Life Science and Technology, ShanghaiTech University, 201210 Shanghai, China. [3] University of the Chinese Academy of Sciences, 100049 Beijing, China. [4] MOE Key Laboratory for Developmental Genes and Human Disease, Institute of Life Sciences, Jiangsu Province High-Tech Key Laboratory for Bio-Medical Research, Southeast University, 210096 Nanjing, China. [5] Co-innovation Center of Neuroregeneration, Jiangsu Key Laboratory of Neuroregeneration, Nantong University, Nantong, China. [6] ENT Institute and Otorhinolaryngology Department of Affiliated Eye and ENT Hospital, Key Laboratory of Hearing Medicine of NHFPC, Shanghai Engineering Research Centre of Cochlear Implant, State Key Laboratory of Medical Neurobiology, Fudan University, 200031 Shanghai, China. [7] School of Physical Science and Technology, ShanghaiTech University, 201210 Shanghai, China. [8] Shanghai Institute of Ceramics, Chinese Academy of Sciences, 200050 Shanghai, China. [9] Department of Otolaryngology Head and Neck Surgery, Affiliated Drum Tower Hospital of Nanjing University Medical School, Jiangsu Provincial Key Medical Discipline (Laboratory), No. 321 Zhongshan Road, 210008 Nanjing, China. [10] Research Institute of Otolaryngology, No. 321 Zhongshan Road, 210008 Nanjing, China. [11] Precise Genome Engineering Center, School of Life Sciences, Guangzhou University, 510006 Guangzhou, China. [12] Institute for Stem Cell and Regeneration, Chinese Academy of Science, Beijing, China. [13] Beijing Key Laboratory of Neural Regeneration and Repair, Capital Medical University, 100069 Beijing, China. [14] These authors contributed equally: Fangzhi Tan, Cenfeng Chu, Jieyu Qi, Wenyan Li. [15] These authors jointly supervised: Huawei Li, Renjie Chai, Guisheng Zhong. Correspondence and requests for materials should be addressed to F.T. (email: tanfzh@shanghaitech.edu.cn) or to H.L. (email: hwli@shmu.edu.cn) or to R.C. (email: renjiec@seu.edu.cn) or to G.Z. (email: zhongsh@shanghaitech.edu.cn)

Hearing loss, one of the most common sensory disabilities, affects over 6.8% of the world's population (~500 million people)[1]. The most current therapy for hearing loss is using hearing devices, which can amplify the sound, enhance sound transmission, or directly stimulate the neurons[2]. This approach currently is the best choice for the treatment of hearing loss, but, unfortunately, still has limitations in hearing sensitivity and perception of natural sound in the noisy environment[2]. Thus, better alternative strategies are required for the treatment of hearing loss, and there are strong clinical needs for the development of biological treatments for the restoration of auditory function[3–5]. Recently, gene therapy has emerged as a possible and promising strategy for the treatment of inherited disease. American Food and Drug Administration (FDA) has approved the first adeno-associated virus (AAV)-mediated gene therapy for patients with rare inherited eye disease in 2017[6], which shed light on the potential gene therapy treatment of hearing loss.

Both hair cells (HCs) and supporting cells (SCs) in the cochlea can be targeted for gene therapy. There are two types of sensory HCs: outer HCs (OHCs), which amplify sound, and inner HCs (IHCs), which convert sound waves to electrical signals[7]. SCs are located between the inner and outer HCs and anchor the sensory epithelium to the basilar lamina, protecting HCs and maintaining its environment. Half of the cases of sensorineural hearing loss are due to genetic mutations in HCs and SCs[8]. Some key deafness genes mainly express and have functions in the SCs[9], such as GJB2, which affects the SCs' gap junction and is the most common hereditary deafness gene[10,11].

In mammals, non-inherited sensorineural hearing loss is mainly due to noise damage, ototoxic drugs, or age-induced HC loss. Although damage to HCs is considered permanent in mammals since they have a very limited capacity for HC regeneration[12], recent studies have shown that SCs are promising inner ear progenitors from which HCs can be regenerated[13]. Thus, SCs are important potential targets for gene therapy, not only for correcting genetic hearing defects but also for HC regeneration.

AAVs have shown clinical efficacy and safety in several organs, including the liver[14] and retina[15]. Nevertheless, to date, there is no cochlear gene therapy based on AAV vectors. Several studies have tested AAV-mediated gene therapies for cochlea in animal models with promising results[16–21]; however, most focused on HCs and spiral ganglion neurons (SGNs)[13]. While the AAV1 serotype was used to deliver wild-type GJB2 into the cochlear SCs of the GJB2-knockout mouse, it could not restore hearing function, likely due to the low transduction efficiency of AAV1 in SCs[22]. Indeed, some AAV serotypes, including AAV1, have been evaluated in the cochlea, and all those tested showed low transduction ratio in SCs (0–13%)[23–25]. Thus, designing novel AAV serotypes that can efficiently infect SCs is key for the development of gene therapy for the treatment of inherited hearing loss and HC regeneration.

Here, we design an AAV variant named AAV-ie (AAV-inner ear) for gene delivery in the mouse cochlea. Ex vivo transduction of mouse organotypic explants shows that AAV-ie can infect nearly 90% of the SCs. In vivo experiments show that after round window membrane (RWM) injection of these AAVs into the cochlea, AAV-ie transduces SCs with significantly higher efficiency than other AAV serotypes. RWM injection of AAV-ie is safe, as indicated by well-preserved HCs and hearing function. Further, after delivering Atoh1 gene into mouse cochlea by AAV-ie, many new HCs are generated, indicating the potential of the AAV-ie vector for HC regeneration.

## Results

**AAV-ie efficiently transduced cochlear SCs.** Previous reports have demonstrated that AAV vectors can be successfully delivered to the inner ear by injection through RWM[26]. With this method, the AAV needs to cross a mesothelial cell layer to infect the HCs and SCs. Thus, novel AAV vectors with the ability to cross the mesothelial cell layer may increase gene transfer efficiency. To this end, we inserted three different cell-penetrating peptides (CPPs) and a CPP-like peptide (DGTLAVPFK) into the VP1 capsid of AAV-DJ (Supplementary Fig. 1a), a synthetic serotype with higher transduction efficiency in cell lines in comparison to any other wild-type serotypes[27]. The peptide (DGTLAVPFK) is from the PHP.eB vector[28]. Its insertion can help the vector cross the blood–brain barrier[28] and thus it shows CPP-like property. Among the tested peptides, only the CPP-like peptide (DGTLAVPFK in the AAV-ie variant) insertion did not decrease the titer of the virus and displayed high infection efficiency in HEK 293T cells (Supplementary Fig. 1b, c).

We then tested whether AAV-ie with its high HEK 293T cell transducing efficiency shows superior properties over other AAV serotypes in organotypic cochlear explants from C57BL/6 mice. Several conventional AAVs, as well as the AAV-ie vector, were used to package a single-stranded (ss) AAV reporter genome that expresses a nuclear mNeonGreen (NLS-mNeonGreen) from the constitutive CAG promoter (ssAAV-CAG-NLS-mNeonGreen). The purified AAV1, 6, 9, Anc80L65 (a synthetic AAV vector that can target OHCs at high rates[25]), AAV-DJ, and AAV-ie were incubated with P4 organotypic cochlear explants. AAV1, 6, 9 and Anc80L65 transduced Sox2-positive SCs at lower efficiencies (1, 11, 1, and 10%, respectively) (Supplementary Fig. 2), while AAV-DJ transduced SCs at a high efficiency of 74% (Supplementary Fig. 2). Interestingly, AAV-ie targeted 90% of the SCs (Supplementary Fig. 2).

Next, we evaluated the SC transduction efficiency of AAV1, 6, 8, 9, PHP.eB, Anc80L65, AAV-DJ, and AAV-ie at equal doses ($3.6 \times 10^9$ genome-containing (particles) (GCs) per ear) in vivo. These vectors were injected into C57BL/6 mice via RWM at P3 and the cochleae were harvested at P14. Consistent with prior reports[23,25], AAV1, 6, 8, 9 and PHP.eB transduced SCs with low efficiencies of <20% (Fig. 1a, b) at the apical region. Anc80L65 and AAV-DJ infected SCs at moderate efficiencies of <55% (Fig. 1a, b). Importantly, AAV-ie transduced the SCs at very high efficiency of 77% (Fig. 1a, b), suggesting that the peptide (DGTLAVPFK) incorporation may help with the crossing of the membrane-like structure and thus increase transduction efficiency. Other cell types were also efficiently targeted by AAV-ie, such as Myo7a-positive HCs and NeuN-positive SGNs (Supplementary Fig. 3). Moreover, delivery of AAV-ie to adult mice showed that it infected cochlear HCs (Supplementary Fig. 4). Taken together, our results indicate that AAV-ie is a powerful viral vector that is capable of infecting not only SCs but also other cell types in the inner ear.

**AAV-ie infects all five types of cochlear SCs.** Although AAV-ie targeted the SCs at the apical region with high efficiency (Fig. 1), we found that only about 30% of SCs at the basal turn were infected by the virus at the dose of $3.6 \times 10^9$ GCs per ear (Supplementary Fig. 5). To increase the targeting efficiency at the basal turn, we injected a high dose of AAV-ie ($1 \times 10^{10}$ GCs) into the cochlea at P3 and harvested the cochlea at P17. Fluorescence microscope experiments revealed that targeting efficiencies of high-dose AAV-ie in cochlear SCs at the apical, middle, and basal regions were about 79%, 83%, and 75%, respectively (Fig. 2a, b), suggesting that AAV-ie infected SCs in a dose-dependent manner. We also harvested the cochlea at P30 after injecting the high-dose AAV-ie at P3. The SCs targeting efficiencies at the apical, middle, and basal regions were about 81%, 77%, and 62%, respectively (Supplementary Fig. 6a, b). In addition, most OHCs

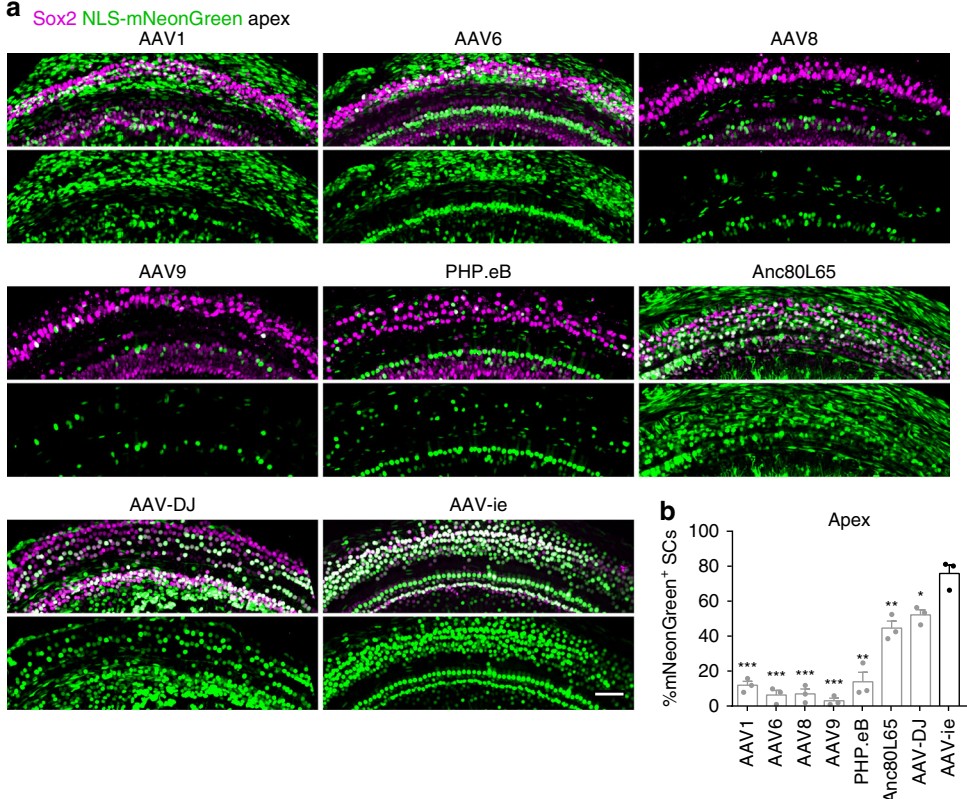

**Fig. 1** Adeno-associated virus-inner ear (AAV-ie) infects cochlear supporting cells with high efficiency. **a** Representative images of nuclear localization sequence-mNeonGreen (NLS-mNeonGreen) fluorescence (green) and Sox2 staining (magenta) in the apical turns of cochleae infected with different AAVs serotypes at the same dose ($3.6 \times 10^9$ genome-containing (particles) (GCs) per ear). Cochleae were harvested at P14 after microinjection with 1.5 μL of AAV stock solution in one ear at P3. mNeonGreen, a green fluorescent protein. Sox2, a marker for supporting cells. Scale bar, 50 μm. **b** Percentage of NLS-mNeonGreen-positive supporting cells per 100 μm corresponding to **a**. Data are shown as mean ± SEM. Significance tests were performed between AAV-ie and other AAV serotypes (see Methods). *p* Value is calculated by Student's *t* test. \**p* < 0.05, \*\**p* < 0.01, and \*\*\**p* < 0.001. *N* = 3 mice. Source data are provided as a Source Data file

and almost all IHCs showed the expression of mNeonGreen at P30 in AAV-ie-injected mice (Supplementary Fig. 6c, d). These results suggest that AAV-ie-mediated gene transfer can achieve long-term expression in the cochlea.

Cochlear SCs contain different cell types: Hensen's cells (HeCs), Deiters cells (DCs), pillar cells, inner phalangeal cells (IPhCs), and inner border cells (IBCs). We found that high-dose AAV-ie infected all cell types of SCs with high efficiency (Fig. 2c–e). In contrast, AAV-DJ at equal dose ($1 \times 10^{10}$ GCs) only targeted inner pillar cells (IPCs), IPhCs, and IBCs with high efficiency (Fig. 2e). All cell types of SCs were infected by Anc80 with low to moderate efficiency (Fig. 2e). The cochlear HCs in adult mice were been infected by AAV-ie (Supplementary Fig. 4), but some of the HCs were missing, likely caused by RWM injection. More importantly, AAV-ie through RWM injection displayed high transducing efficiency in cochlear SCs in adult mice (Fig. 2f). The SCs targeting efficiencies at the apical, middle, and basal regions were about 62%, 56%, and 54%, respectively (Fig. 2g). Thus, these results suggest that AAV-ie is more suitable for targeting cochlear SCs, in comparison with other AAVs.

**AAV-ie infects vestibular epithelia with high efficiency**. Since perilymphatic solutions of the cochlea are continuous with those of the vestibular labyrinth, the virus can diffuse to the vestibular sensory organs after RWM injection. Therefore, we tested whether AAV-ie injection at P3 would transduce the vestibular organs, which are responsible for detecting linear motion and

sensing gravity. We found strong NLS-mNeonGreen expression throughout the sensory epithelium of mouse utricle (Fig. 3a, c). AAV-ie successfully infected almost 100% of the HCs and SCs (Fig. 3b, d). In addition, delivery of AAV-ie at P30 showed that it also infected the adult utricle (Supplementary Fig. 7).

To test the possibility that AAV-ie may be a useful vector for the treatment of human vestibular dysfunction, we harvested human vestibular epithelia from one adult patient and cultured the epithelium. AAV-ie-transduced samples showed strong mNeonGreen fluorescence in both utricular SCs and HCs (Fig. 3e), and about 93% of utricular SCs and 76% of HCs could be transduced by AAV-ie. Moreover, we also tested the targeting efficiencies of AAV-ie in human saccule and crista and found that AAV-ie can infect the HCs and SCs with moderate to high efficiencies (Supplementary Fig. 8). Thus, these data indicate that AAV-ie may be a useful vector for the clinical treatment of vestibular dysfunction.

**AAV-ie is safe for use in mouse inner ear**. In order for AAV to be a viable tool for the treatment of hearing loss, the AAV used should have minimal effect on the auditory system. To assess HC survival after AAV-ie infection, we injected the virus into the cochlea at P3 and performed scanning electron microscopy (SEM) analysis along the organ of Corti at P30. Fluorescence microscope experiments revealed the strong NLS-mNeonGreen expression throughout the cochlea at P30 (Fig. 4a). The SEM results showed that OHCs and IHCs of AAV-ie-injected mice

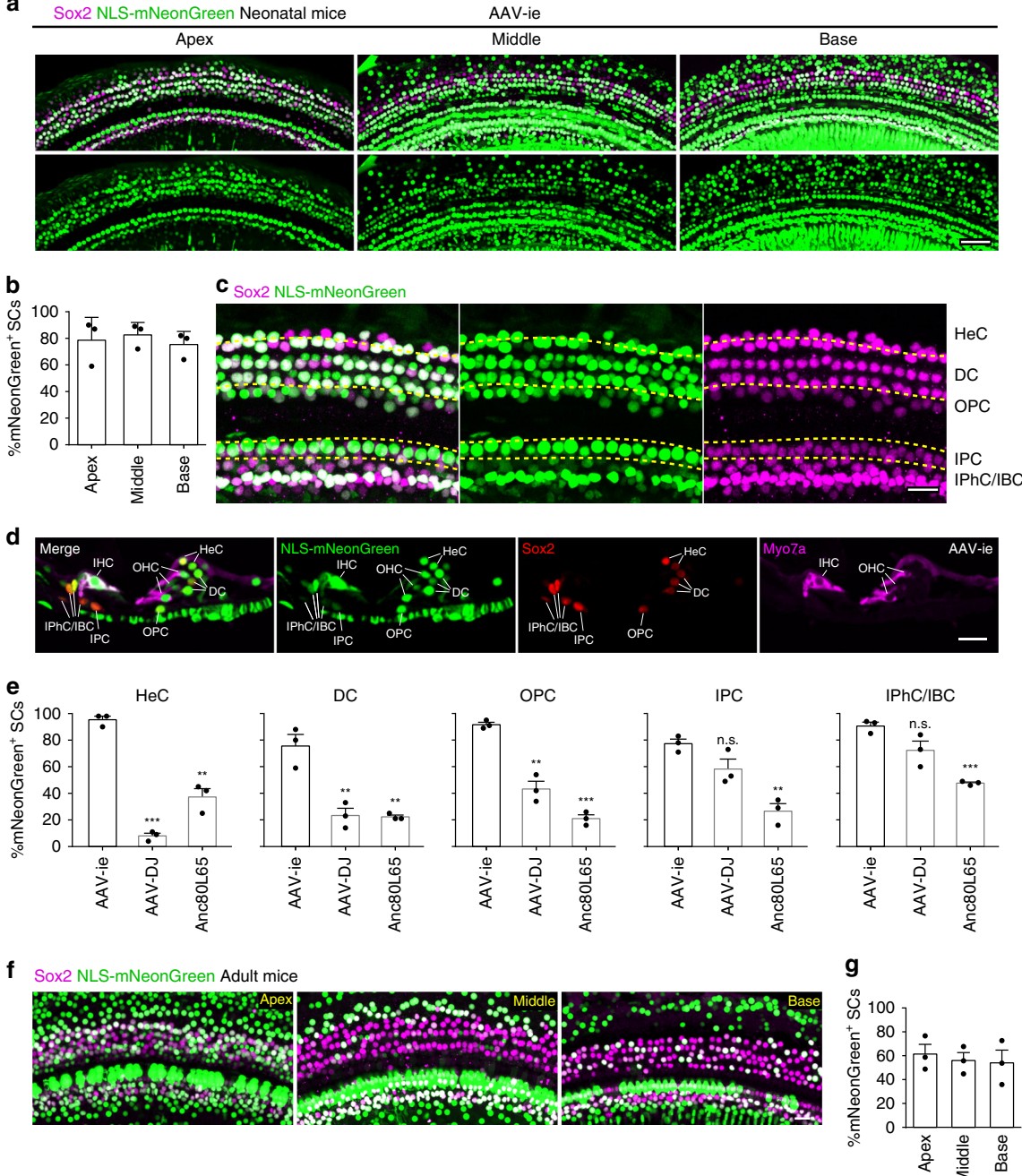

**Fig. 2** Adeno-associated virus-inner ear (AAV-ie) efficiently targets cochlear supporting cells in neonatal and adult mice. **a** Images of the apical, middle, and basal turns of cochlea injected at P3 with AAV-ie-NLS-mNeonGreen (nuclear localization sequence-mNeonGreen) at a dose of $1 \times 10^{10}$ genome-containing (particles) (GCs) per ear. Green: NLS-mNeonGreen; magenta: Sox2. Scale bar, 50 μm. **b** Percentage of NLS-mNeonGreen-positive supporting cells (SCs) per 100 μm corresponding to **a**. Data are shown as mean ± SEM. $N = 3$ mice. Source data are provided as a Source Data file. **c** Confocal images of Hensen's cells (HeCs), Deiters cell (DCs), outer pillar cells (OPCs), inner pillar cells (IPCs), inner phalangeal cell (IPhCs), and inner border cell (IBCs) in apical turn of cochleae injected with AAV-ie-NLS-mNeonGreen. Green: NLS-mNeonGreen; magenta: Sox2. Scale bar, 20 μm. **d** Cross-section of AAV-ie-NLS-mNeonGreen injected cochlea ($1 \times 10^{10}$ GCs per ear). Green: NLS-mNeonGreen; red: Sox2; magenta: Myo7a (a marker for hair cells). Scale bar, 20 μm. **e** Infection efficiencies of Anc80L65, AAV-DJ, and AAV-ie at an equal dose ($1 \times 10^{10}$ GCs) in HeCs, DCs, OPCs, IPCs, IPhC/IBCs, respectively, are quantified. Data are shown as mean ± SEM. $p$ Value is calculated by Student's $t$ test. n.s. refers to no significance. $*p < 0.05$, $**p < 0.01$, and $***p < 0.001$. $N = 3$ mice. Source data are provided as a Source Data file. **f** Representative confocal images of cochlear supporting cells of adult mice injected with AAV-ie-NLS-mNeonGreen. Green: NLS-mNeonGreen; magenta: Sox2. Scale bar, 20 μm. **g** Infection efficiencies of AAV-ie-infected SCs in adult mice. Data are shown as mean ± SEM. $N = 3$ mice. Source data are provided as a Source Data file. For neonatal mice (**a–e**), all cochleae were harvested at day 14 after microinjection with 1.5 μL of AAV stock solution at P3. While for adult mice (**f**, **g**), all cochleae were harvested 14 days after microinjection at P30

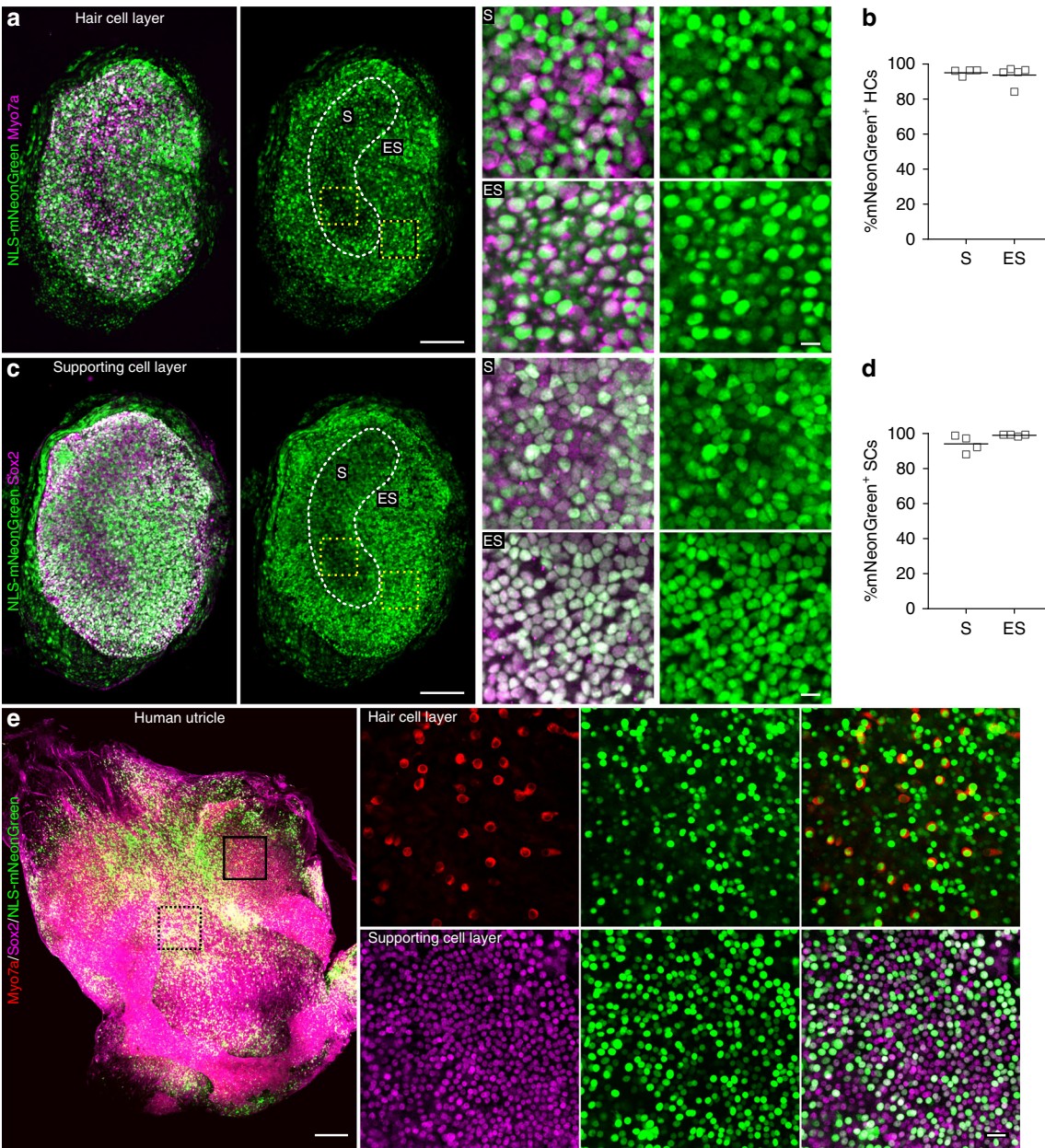

**Fig. 3** Adeno-associated virus-inner ear (AAV-ie) transduction in vestibular sensory epithelia. **a–d** Mouse utricle from a P3 mouse injected with AAV-ie-NLS-mNeonGreen (nuclear localization sequence-mNeonGreen) ($1 \times 10^{10}$ genome-containing (particles) (GCs) per ear). The tissues were harvested at P17 fixed and stained with anti-Myo7a antibody (**a**, magenta) or anti-Sox2 antibody (**c**, magenta) and imaged for mNeonGreen fluorescence (green). **a**, Left, representative confocal images of stained mouse utricular hair cells. Scale bar, 100 μm. Right, magnified regions from left yellow boxes. Scale bar, 10 μm. Green: NLS-mNeonGreen; magenta: Myo7a. S and ES indicate the striolar region and extrastriolar region, respectively. **b** Percentage of NLS-mNeonGreen-positive utricular hair cells in S and ES regions. Horizontal bars represent mean values. $N = 5$ mice. Source data are provided as a Source Data file. **c**, **d** Same as **a**, **b** except that utricular supporting cells were captured and analyzed. Horizontal bars represent mean values. $N = 4$ mice. Source data are provided as a Source Data file. **e**, Left, The sensory epithelium of an adult human utricle. The tissue was exposed to $5 \times 10^{10}$ GCs AAV-ie-NLS-mNeonGreen for 24 h, cultured for 7 days, fixed, stained with anti-Myo7a antibody (red) or anti-Sox2 antibody (magenta), and imaged for mNeonGreen fluorescence (green). Scale bar, 250 μm. Right, magnified regions from human utricle in hair cell layer and supporting cell layer. Scale bar, 20 μm

were preserved (Fig. 4b) and their hair bundles were properly oriented (Fig. 4c). We then estimated HC counts in representative fields of view. The HC numbers between uninjected and injected ear showed no difference (Fig. 4d). Thus, these data suggest that AAV-ie has little or no toxic effect on HCs. To investigate systems-level hearing function after injection of AAV-ie, auditory brain-stem responses (ABRs) from three AAV-ie-injected ears and three uninjected ears were measured. ABR thresholds showed no difference between injected and uninjected ears (Fig. 4e),

suggesting that this vector had no negative effect on auditory function.

**In vivo HCs regeneration mediated by AAV-ie-*Atoh1*.** Manipulation of signaling pathways and transcription factors such as gene *Atoh1* can lead to transdifferentiation of SC into HCs[29]. To assess the potential of the AAV-ie vector for HC regeneration, we used AAV-ie-*Atoh1*-NLS-mNeonGreen (AAV-ie-*Atoh1*) to

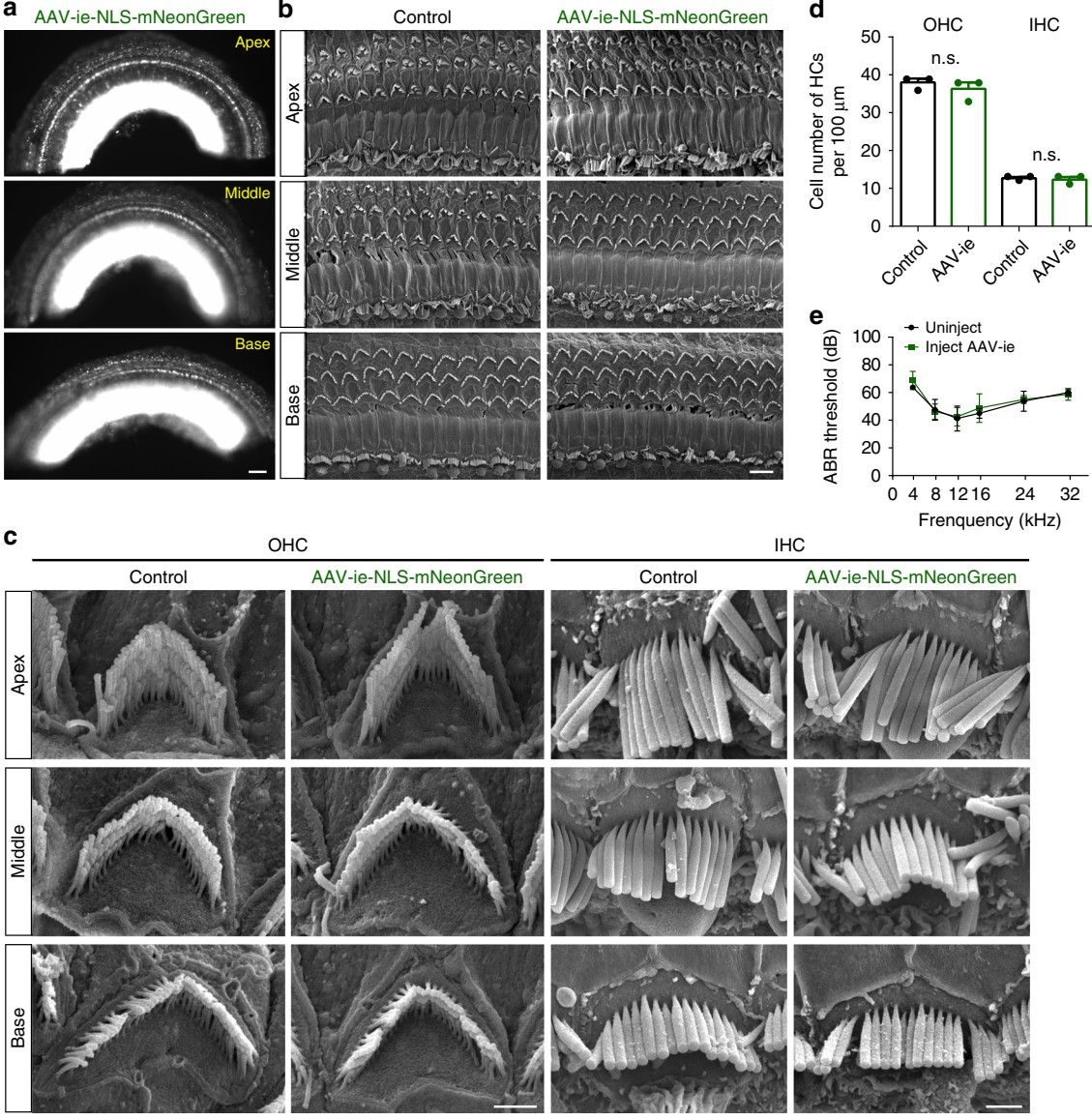

**Fig. 4** Adeno-associated virus-inner ear (AAV-ie) has a minimal adverse effect on auditory in injected mice. **a** Representative epi-fluorescence image of P30 AAV-ie-NLS-mNeonGreen (nuclear localization sequence-mNeonGreen)-injected cochlea. The mice were injected at P3 at a dose of 1 × 10^10 genome-containing (particles) (GCs) per ear. Scale bar, 100 μm. **b** Scanning electron microscopy (SEM) images of P30 control and AAV-ie-NLS-mNeonGreen-injected cochlea at apical, middle, and basal regions. Scale bar, 5 μm. **c** Magnification SEM images of outer hair cell (OHCs) and inner hair cells (IHCs) of P30 control and AAV-ie-NLS-mNeonGreen-injected cochlea at apical, middle, and basal regions. Scale bar, 1 μm. **d** OHCs and IHCs numbers per 100 μm of P30 control and AAV-ie-NLS-mNeonGreen-injected cochlea. $N = 3$ mice. $p$ Value is calculated by Student's $t$ test. n.s. refers to no significance. Source data are provided as a Source Data file. **e** Auditory brain-stem response (ABR) thresholds from AAV-ie-NLS-mNeonGreen-injected and -uninjected ears on day 26 after surgery at P3. Error bars ± SEM. $N = 4$ mice, respectively. Source data are provided as a Source Data file

deliver mouse *Atoh1* into cochlea. The AAV-ie-*Atoh1* virus (1 × 10^10 GCs per ear) was injected into C57BL/6 mice via RWM at P0 and the cochleae were harvested at P14. In the control group, there were no Myo7a⁺ ectopic HCs in any area of the cochlea (Fig. 5a, left panel). In the AAV-ie-*Atoh1* group, a large number of new HCs expressing Myo7a were present in the sensory region as well as in the greater epithelial ridge region (Fig. 5a, right panel). Some new HCs retained the expression of Sox2, a cochlear SC-specific marker, indicating that there are differences in the extent of conversion among new HCs (Fig. 5b, white arrows). The total number of new HCs in the sensory region was as high as 82 per 100 μm (Fig. 5c). Of note, we observed little difference in the number of new HCs in the AAV-ie-*Atoh1* group as compared to a previous study that used *Foxg1*-Cre-mediated *Atoh1*

overexpression mice[30], which suggests that AAV-ie is a powerful viral vector for HC regeneration.

We next assessed the cell properties of the new HCs. Fluorescence imaging with the phalloidin labeling clearly showed hair bundles in new HCs induced by AAV-ie-*Atoh1*, a feature that SCs do not possess (Fig. 5d). Then, we performed SEM experiments, which revealed hair bundles along the organ of Corti, an HC-like feature (Fig. 5e). Both experimental results identify the hair bundle feature of newly generated cells induced by AAV-ie-*Atoh1* (Fig. 5d, e). To further characterize the property of new HCs induced by AAV-ie-*Atoh1*, we recorded the membrane properties with electrophysiological experiments. As expected, SCs near the IHCs did not generate action potentials (Fig. 5f), while IHCs can produce action potentials upon current

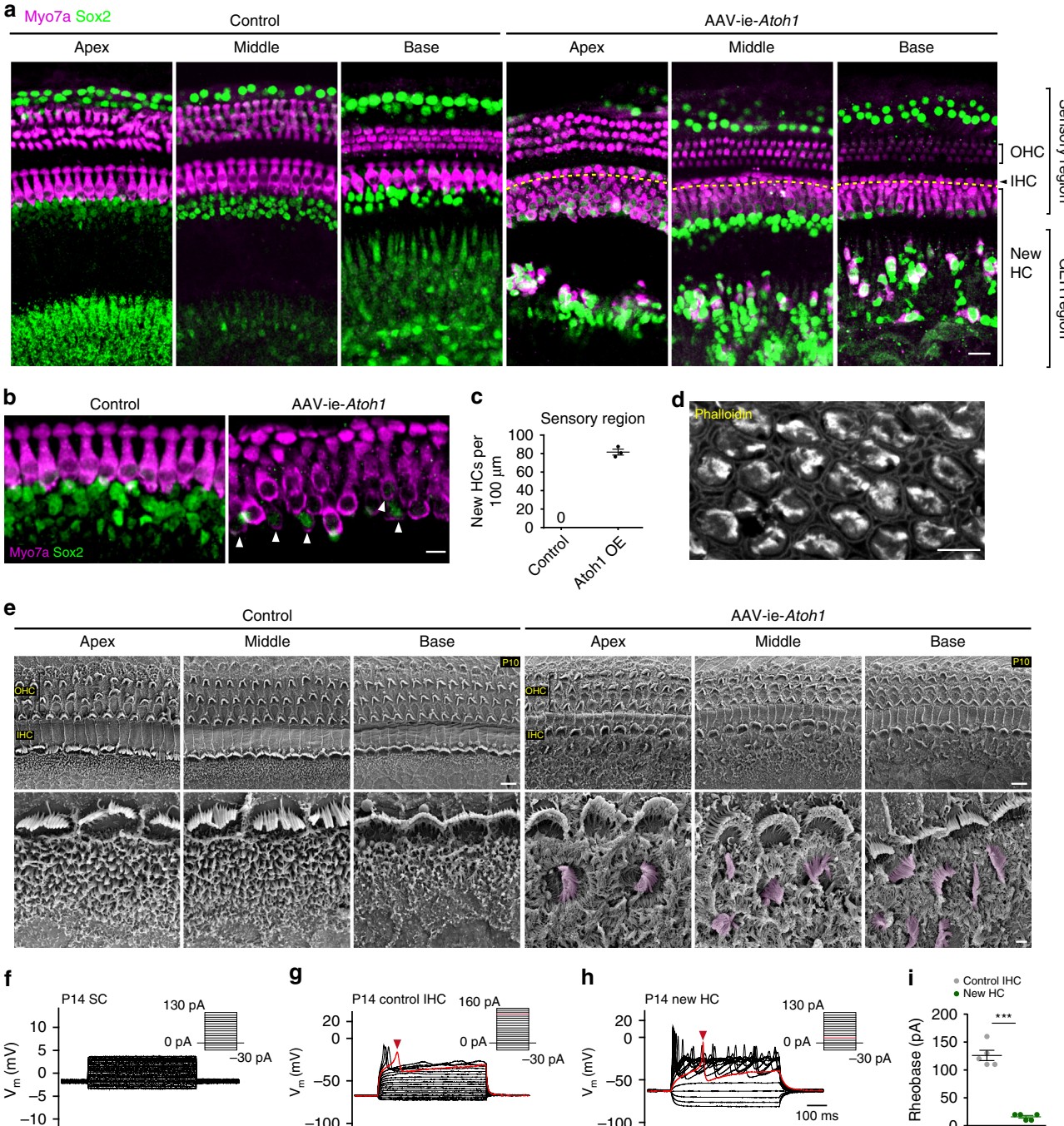

**Fig. 5** Adeno-associated virus-inner ear-*Atoh1* (AAV-ie-*Atoh1*) induces new hair cells (HCs) in vivo with stereocilia. Mice were injected with AAV-ie-*Atoh1* ($1 \times 10^{10}$ genome-containing (particles) (GCs)) at P0 and the cochlea was harvested at P14. **a** Representative confocal projection image of control and AAV-ie-*Atoh1* cochlea. Scale bar, 20 μm. **b** Magnification of inner HC (IHC) region of control and AAV-ie-*Atoh1* cochlea. White arrows indicate both Sox2- and Myo7a-positive new HCs. Green: Sox2; magenta: Myo7a. Scale bar, 10 μm. **c** Number of Myo7a-positive new HCs per 100 μm in sensory region. Data are shown as mean ± SEM. $N = 3$. Source data are provided as a Source Data file. **d** Representative confocal image of phalloidin staining of new HCs in AAV-ie-*Atoh1* cochlea. Scale bar, 10 μm. **e** Scanning electron microscopy (SEM) images of AAV-ie- and AAV-ie-*Atoh1*-injected cochlea at apical, middle, and basal regions. Regenerated HC-like cells were artificially colored magenta. Scale bars, 5 μm (upper panels), 1 μm (lower panels). **f** Representative membrane responses of P14 supporting cells (SCs) to current. The trace shows action potential generation in response to 10 pA injections. $N = 5$. **g** Same as **f**, but for P14 IHCs. The trace shows action potential generation in response to 10 pA injections. The first action potential was generated by 130 pA injection (red arrow). **h** Same as **f**, but for P14 new HC injected with AAV-ie-*Atoh1* and the first action potential was generated by 20A injection (red arrow). **i** Average responses show significant difference between IHCs and regenerated new HCs. Data are shown as mean ± SEM. *p* Value is calculated by Student's *t* test. ***$p < 0.001$. $N = 5$. Source data are provided as a Source Data file

stimulations, reflecting the excitatory characteristics (Fig. 5g, red arrow, and Fig. 5i). We hypothesized that new HCs induced by AAV-ie-*Atoh1* may represent a transitional state between SCs and IHCs and may show some excitatory membrane features. Indeed, the new HCs induced by AAV-ie-*Atoh1* were excitatory and did generate action potentials upon current stimulations (Fig. 5h, i). Together, these observations suggest that SCs overexpressing *Atoh1* represent a primitive type of HCs and that AAV-ie represents a power viral vector for regenerating HCs.

## Discussion

Recently, a synthetic AAV Anc80L65 shows the promise to treat hereditary hearing loss by infecting cochlear OHCs and IHCs with high efficiency[25,31]. Another AAV AAV2.7m8 shows higher infection efficiency than Anc80L65, but whether it can improve the auditory function in hereditary hearing loss remains unknown[32]. Here, we developed an AAV variant, AAV-ie, by inserting a peptide, DGTLAVPFK, which may help the vector to cross-membrane-like structure in order to universally increase the infection efficiency. AAV-ie highly transduces the cochlear HCs (Supplementary Fig. 3), and utricular HCs (Fig. 3). Further, AAV-ie has the capability to transduce the vestibular HCs from human clinical samples (Fig. 3e and Supplementary Fig. 8). Importantly, AAV-ie does not affect the function of HCs and auditory system (Fig. 4). Thus, AAV-ie may serve as an efficient tool for delivering genes into mammalian HCs.

Hearing loss is often caused by aging, noise, or other environmental factors. One strategy is to regenerate HCs to restore the auditory function. Unfortunately, adult mammalian HCs cannot be regenerated after damage or stress. Thus, to find effective treatments to regenerate HCs represents a big chance and challenge in the auditory field. Studies show that third row of DCs, IPCs, and IPhCs, which express leucine-rich repeat-containing G-protein coupled receptor 5, are the main SCs that can generate new HCs[33,34]. The IPCs and IPhCs are near IHCs and maybe the progenitors from which IHCs can be regenerated. The DCs are below the OHCs and have the capacity of regenerating OHCs. Recently, AAV2.7m8[32] was found to be a useful vector for targeting cochlear HCs and SCs, but it only infected two types of SCs (IPCs and IPhCs) and barely targeted HeCs, DCs, and OPCs. AAV-ie targets all the cochlear SC cell types in the cochlea and almost 100% SCs in the utricle with high efficiencies (Fig. 2c, d), indicating that AAV-ie is more suitable not only for correcting genetic defects in SCs but also for HC regeneration.

*Atoh1* plays key roles in HC determination and its ectopic expression in SCs induces their transdifferentiation to HCs[13]. Several viral vectors, including lentivirus and adenovirus, have been used to deliver *Atoh1* into the cochlea, and adenovirus-mediated *Atoh1* overexpression has induced the regeneration of HCs and partially restored hearing in deaf guinea pigs[35,36]. Based on these studies, FDA approved a human clinical trial using an adenovirus vector to deliver human *Atoh1* gene into the cochlea (NCT02132130). However, the transduction and regeneration efficiencies of adenovirus need to be improved[35,36]. Here, our results show that AAV-ie-*Atoh1* can induce many new HCs in the sensory region (Fig. 5) and the efficiency is comparable with previous work, which used mice with specific Cre-mediated *Atoh1* overexpression in SCs[30]. Considering the regeneration efficiency and the strong immune response of adenovirus, AAV-ie has a higher potential for HC regeneration. However, we noted that the newly generated HCs were relatively immature (Fig. 5d–i). This is consistent with previous studies[37,38], which showed that *Atoh1* only was not sufficient to regenerate mature HCs. Future study will be required to identify better ways to regenerate fully functional HCs.

Except for SCs, most HCs and SGNs were transduced by AAV-ie (Supplementary Fig. 3). Considering the unpredictable effects of gene transfer to HCs and SGNs as well as other cell types, the broad tropism of AAV-ie in the inner ear may limit the application of AAV-ie for correcting the gene mutation in SCs or HC regeneration. This problem could be solved by screening AAVs, which can specifically transduce SCs or using SC-specific promoters, such as glial fibrillary acidic protein (*GFAP*) promoter. The *GFAP* promoter has been widely used to specifically express proteins in astrocyte[37]. In the cochlea, GFAP protein is expressed in SCs, but not in HCs[38]. Thus, the *GFAP* promoter can help AAV-ie to specifically express genes in SCs.

In conclusion, we identified AAV-ie with the ability to infect cochlear HCs and SCs safely and with high efficiency in mice. Further validation of AAV-ie as a gene transfer vector for clinical use will require studies in adult large-animal models, as well as in animal disease models. Considering that some deafness genes mainly affect SC function and that SCs have the ability to regenerate HCs, AAV-ie may hold significant potential, not only for correcting genetic hearing impairment but also for HC regeneration to treat environmental and age-induced hearing loss.

## Methods

**Animal.** Animals were used in accordance with standard ethical guidelines. C57BL/6 mice of both sexes in an estimated 50:50 ratio were used in this study. The number of days since birth was counted from postnatal day 0 (P0). Animals were housed under a 12 h light/dark cycle at a room temperature of $22 \pm 1$ °C with food and water available ad libitum. All experiments were approved by the Institutional Animal Care and Use Committee of ShanghaiTech University and Southeast University, China, and all efforts were made to minimize the number of mice used and their suffering.

**Virus preparation.** The C-terminal Flag-tagged NLS-mNeonGreen was cloned into the AAV plasmid containing the cytomegalovirus enhancer/chicken $\beta$-actin (CAG) promoter and the woodchuck hepatitis virus post-transcriptional regulatory element (WPRE) cassette that was flanked by AAV2 inverted terminal repeats. The various AAV serotype vectors were produced in HEK 293T cells co-transfected with rep-cap fused plasmid and a helper plasmid. AAVs were purified by iodixanol gradient ultracentrifugation[39]. In brief, cultured medium was collected twice every 48 h after transfection. The cell lysate was treated with chloroform and the supernatant was collected. The medium and the supernatant were combined and concentrated by precipitation with 10% PEG 8000 and 1.0 M NaCl. After centrifugation, the pellet was resuspended in PBS buffer with Benzonase. 15%, 25%, 40%, and 60% iodixanol solutions were carefully layered and then the viral suspension generated was overlaid. Then centrifuged at 350,000 g for 90 min at 10 °C. Following ultracentrifugation, the AAV-containing 40% fraction was collected. The buffer was exchanged to remove the iodixanol and concentrated the purified virus. The genome-containing titers of AAVs were determined by SYBR (Roche) analysis using primers targeting the WPRE region. The qPCR primers for WPRE are listed as follows: forward, 5′-GTCAGGCAACGT GGCGTGGTGTG-3′; reverse, 5′-GGCGATGAGTTCCGCCGTGGC-3′.

**In vitro explant culture.** We followed procedures as published previously[25]. In brief, the P3 wild-type C57BL/6 mouse cochleae were quickly removed from the temporal bone. The tectorial membrane was peeled off and attached to a Cell-Tak-coated slide and placed in 98% Dulbecco's modified Eagle's medium (DMEM) + 1% N2 + 1% Amp (5 μg mL$^{-1}$) medium for 12 h. Then, 1% fetal bovine serum and $2 \times 10^{10}$ GC AAV was added to the culture medium for 60 h. Two specimens were obtained per cochlea: the lower apical turn and the upper basal turn for each type of virus.

The human vestibular sensory organs were collected from patients by labyrinthectomy surgery due to the petrous apex cholesteatoma or the acoustic neuroma, including the maculae utriculi, maculae sacculi, and three crista ampullaris. The freshly isolated tissues were kept in culture medium, the advanced DMEM/F12 supplemented with penicillin/streptomycin (2%, Sigma), N2 (1%, Thermo Fisher Scientific), B27 (2%, Thermo Fisher Scientific), epidermal growth factor (20 ng mL$^{-1}$; Thermo Fisher Scientific), basic fibroblast growth factor (10 ng mL$^{-1}$; Thermo Fisher Scientific) and insulin-like growth factor (50 ng mL$^{-1}$; Thermo Fisher Scientific), at 37 °C with 5% $CO_2$ in 4-well Petri dishes (Greiner Bio-One). After 12 h culture, the sensory organs were exposed to $5 \times 10^{10}$ GC AAV for 24 h and maintained in culture for 7 days. This project was approved by the Research Ethics Committee of the Institution of Eye and ENT Hospital of Fudan University.

**AAV injection by round window injection**. We followed surgery as published previously[25]. Briefly, the neonatal mice were anesthetized by low temperature. P2–3 mice were placed in an ice bath for 2–3 min until loss of consciousness and then removed to an ice pad for subsequent surgical procedures. The surgery was limited to 5–10 min. Surgery was performed only on the left ear of each animal. The right ear served as a negative control. Upon anesthesia, a post-auricular incision was made to expose the otic bulla and visualize the cochlea. According to the relative position between the temporal bone and the facial nerve, the round window was exposed. Special care was taken to avoid damage to the facial nerve during surgery. Injections were performed through the RWM with a glass micropipette (25 μm) controlled by a micromanipulator UMP3 UltraMicroPump (World Precision Instruments). The volume of the injected materials was controlled at ~1.5 μL per cochlea within 1 min per AAV. After the injection, the skin incision was closed using veterinary tissue adhesive (Millpledge Ltd, UK). Pups were subsequently returned to the 38 °C warming pad for 10 min and then returned to their mother for continued nursing. A glass micropipette was inserted so as to enable a visual inspection of any damage to the exposed structures of the ear. Any damaged cochlea were excluded from subsequent studies. Success rates of injection ranged between ~50 and ~70% depending on the proficiency of the researcher.

For RWM injection in adult mice, 4-week-old C57/B6 mice were used. Before surgery, the mice were anesthetized with ketamine (100 mg kg$^{-1}$) and xylazine (25 mg kg$^{-1}$). The otic bulla was opened to expose the round window niche. Injections were performed through the RWM with a glass micropipette (25 μm) controlled by a micromanipulator UMP3 UltraMicroPump (World Precision Instruments). The volume of the injected materials was controlled at approximately 2.0 μL per cochlea within 1 min per AAV. After the injection, the skin incision was closed using veterinary tissue adhesive (Millpledge Ltd, UK).

**Hearing tests via ABR measurements**. ABR is a method to assess hearing by measuring the hearing threshold. Based on the Tucker-Davis Technology system III (Tucker-Davies Technologies, TDT, Gainesville, FL, USA), in this test closed-field ABR was recorded from mice anesthetized with ketamine (100 mg kg$^{-1}$) and xylazine (25 mg kg$^{-1}$) in a sound-attenuated room and changes in the electrical activity of the brain in response to sound were recorded via electrodes. The ABR responses were elicited in tone bursts at five frequencies (4, 8, 12, 16, 24, and 32 kHz). The acquired response signal of ABR was amplified (10,000 times), filtered (0.1–3 kHz), averaged, and presented in a computer-based data-acquisition system, BioSigRZ software (TDT, Gainesville, FL, USA). Sound level was raised in 5 to 10 dB steps from 0 to 90 dB sound pressure level (decibels SPL). At each level, 1024 responses were averaged (with alternating stimulus polarity) after "artifact rejection." The threshold was defined as the lowest sound pressure level that elicited a visually detectable response. Three AAV-ie-injected ears and three uninjected ears were analyzed at P28–P30.

**Immunofluorescence staining**. Immunostainings were performed on both cultured and injected samples. For cultured cochlea or injected utricle, samples were immersed in 4% paraformaldehyde (PFA) for fixed processing before blocking. Injected cochleae were harvest at day 14 after injection. The temporal bone was fixed in 4% paraformaldehyde (in phosphate-buffered saline (PBS), pH 7.2) for 2 h at room temperature and was then cut into pieces after softening for 0.5–6 h after treatment with 0.5 mM EDTA (pH 8.0). For SGN immunostaining, serial cryostat sectioning of the temporal bone embedded in OCT was performed after gradient dehydration in sucrose solutions (15% sucrose, 30% sucrose, 30% sucrose: OCT = 1:1). For cross-section, cochleae were excised and fixed in 4% PFA for 2 h and then decalcified in 0.5 M EDTA for 2 h at room temperature. For frozen sections, specimens were cryo-protected in 30% sucrose in PBS overnight at 4 °C and then embedded in OCT compound, frozen, and sectioned (14 μm).

Samples were blocked with 10% donkey serum in 0.3% Triton X-100 dissolved in PBS at room temperature for 1 h. Then, the samples were stained with antibodies against myosin 7A (Myo7a, #25-6790 Proteus Biosciences, 1:1000), Sox2 (Sox2, #sc-17320, Santa Cruz Biotechnology, 1:1000), Flag (Flag, #F3165, Sigma-Aldrich, 1:1000), and NeuN (NeuN, #12943S, Cell Signaling Technology, 1:500) together with corresponding secondary antibodies. Samples were mounted with anti-fluorescence quenching agent Vectorshield (Vectorlabs) mounting media and confocal microscopy was used for observation.

**Scanning electron microscopy**. Temporal bones were fixed in 2.5% glutaraldehyde at 4 °C overnight, and then were rinsed in 0.1 M PBS. Temporal bones were decalcified in 0.5 M EDTA (pH 8) at room temperature for 2–4 h, and then the organ of Corti was dissected into apical, middle, and basal regions. Specimens were rinsed in 0.1 M PBS and post-fixed in 2% osmium tetroxide for 3 h at 4 °C. Samples were dehydrated through a graded ethanol series 10 min for each step: 30%, 50%, 70% 90%, and 100%. Samples were emplaced to conductive cloth tape after being critical point dried (SAMDRI-795, Tousimis Ltd.). Cochleae were sputter coated with 10 nm of platinum (JEC-3000FC, JEOL Ltd.), and observed with a field emission SEM (JSM-7800F prime, JEOL Ltd.).

**Whole-cell patch-clamp recording**. Cochlear specimens were prepared from P14 mice of either sex. New HCs, IHCs, or SCs at the apical location were randomly chosen for recordings. Cochleae were dissected in the dissection solution: 5.36 mM KCl, 141.7 mM NaCl, 1 mM MgCl$_2$, 0.5 mM MgSO$_4$, 0.1 mM CaCl$_2$, 10 mM HEPES, 3.4 mM L-glutamine, and 10 mM D-glucose (325 mOsm, pH = 7.25). When recording, cochleae were incubated in the external solution: 144 mM NaCl, 0.7 mM NaH$_2$PO$_4$, 5.8 mM KCl, 1.3 mM CaCl$_2$, 0.9 mM MgCl$_2$, 10 mM HEPES, and 5.6 mM D-glucose (325 mOsm, pH = 7.35). Whole-cell patch-clamp recordings were performed using an Olympus microscope (BX51WI) and data were collected and analyzed using the Axopatch1500B amplifier and pCLAMP10 software (Molecular Devices). Patch pipettes had an impedance of 10–15 MΩ when filled with the internal solution containing 135 mM KCl, 0.1 mM CaCl$_2$, 5 mM MgCl$_2$, 3.5 mM MgCl$_2$, 2.5 mM Na$_2$ATP, and 5 mM HEPES-KOH (325 mOsm, pH = 7.35). For all recordings, the leak currents were subtracted using P/4 procedure. Recordings were low-pass filtered at 2 kHz.

**Quantification of mNeonGreen expression**. For in vitro experiments, samples were immunostained with Myo7a and Sox2, together with Flag antibody to enhance NLS-mNeonGreen signals. For in vivo experiments, samples were immunostained with Myo7a, Sox2, or NeuN. When capturing the images using confocal microscopy, the laser power setting for samples infected by AAV-ie-NLS-mNeonGreen was selected as a standard. All visible NLS-mNeonGreen signals were captured using the same laser setting. For data processing, the percentage of NLS-mNeonGreen-positive HCs and SCs was manually quantified along the cochlea, by counting the number of NLS-mNeonGreen-positive, Myo7a-positive HCs, and Sox2-positive SCs (including DCs, OPCs, and IPCs) per 100-μm sections per apical sample for each specimen. For NLS-mNeonGreen-positive cells counting in vestibular HCs and SCs, images were taken from 1 to 2 representative areas from the striolar and/or extrastriolar regions per utricle for analyses. If needed, Z-stack image projections were performed.

**Statistical analysis**. All data were shown as the mean or mean ± SEM and all experiments were repeated at least three times. Statistical analyses were carried out using the Excel (Microsoft) and GraphPad Prism 6.0 software. ABR results were analyzed with a two-tailed, unpaired Student's $t$ tests. A value of $p < 0.05$ was considered statistically significant. Error bars and $n$ values are defined in the respective figures and legends.

**Reporting summary**. Further information on research design is available in the Nature Research Reporting Summary linked to this article.

## Data availability
The source data underlying Figs. 1–5 and Supplementary Figs. 1–3, 5, and 6 are provided in the Source Data file. All data are available upon reasonable request.

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

## Acknowledgements

We thank the Shanghai Municipal Government and ShanghaiTech University, and the Bioimaging Core Facilities of the iHuman Institute and the animal facility of National Center for Protein Science Shanghai for their financial support. This work was also supported by the Strategic Priority Research Program of the Chinese Academy of Science (XDA16010303), the National Key Research and Development Program of China (2016YFC0905900 (G.Z.), 2017YFC 1001300 (G.Z.), 2015CB965000 (R.C.), 2017YFA0103900 (R.C.)), the National Natural Science Foundation of China (31771130 (G.Z.), 31601185 (F.T), 81622013 (R.C.), 81620108005 (H.L.)), the 2015 Thousand Youth Talents Plan of China (G.Z., R.C.), Boehringer Ingelheim Pharma GmbH (R.C.), and the Yingdong Huo Education Foundation (R.C.). We also thank Qing Zhang, Yilan Jiang, and Yujuan Zhao of Center of High-resolution Electron Microscopy, SPST, and ShanghaiTech University (Grant No. 02161943) for providing technical support and assistance in data collection and analysis.

## Author contributions

G.Z. initiated and managed the project. G.Z., F.T., H.L. and R.C. conceived and designed the experiments. C.C., F.T., J.Q. and W.L. performed most of the experiments and data analysis. C.C., J.Q. and K.L. performed the SEM experiments and data analysis. Electrophysiological recordings and analysis were performed by X.C. All the authors contributed to data analysis, interpretation, and presentation. G.Z. and F.T. wrote the manuscript with contributions from all of the authors.

## Additional information

**Competing interests:** A patent based on this manuscript has been submitted. The patent applicant is ShanghaiTech University. The inventors are Fangzhi Tan, Guisheng Zhong, Cenfeng Chu and Jieyu Qi. The application number is Chinese Patent Application No.201910093000.3. The status of the application is pending. The specific aspect of the manuscript covered in the patent application is "AAV-ie vector and AAV-ie mediated gene delivery in cochlea." The remaining authors declare no competing interests.

