## [Peer Review File · Nature Communications]

Reviewers' Comments:

Reviewer #1:

Remarks to the Author:

In this work, Tan and colleagues have devised various peptide-displaying variants of the previously reported AAV-DJ capsid and then selected one for further validation as a vector for gene transfer to supporting cell in the mouse inner ear.

Overall, I find the data and this work interesting. Unfortunately, however, the way the data are presented makes it impossible for me to truly appreciate the potential value of the AAV-ie vector, for reasons detailed below. Most importantly, I was frequently confused by the changing nomenclature regarding the AAV transduction experiments in which the authors switch back and forth between volumes, doses and titers. In the absence of a unified nomenclature (ideally, vector doses), I cannot fully interpret the shown data and hence cannot scientifically evaluate this work. Fortunately, though, this could be fixed in a revised manuscript provided the editors will make such an invitation. To this end, specific comments and suggestions for improvement are:

On p.3, the authors mention they inserted five different CPPS, yet Fig. S1a shows only four. Besides, why is the PHP.eB peptide (part of the four) classified as CPP; what are the data supporting this?

What was the rationale for pre-screening these variants in HEK293T (kidney) cells and then taking the lead candidates from this screen forward to the studies in the actual target cells? How predictive, if at all, are HEK293T cells for studies in the inner ear? Why not test all vectors directly in the on-target cells? Has this been done, and, if so, what were the results?

At the bottom of page 4, I was confused. In the sentence starting with "However, a large number of ...", are the authors now talking about the vector-treated cohort (since they were referring to the control group in the sentence before)?

In Figure 1, my understanding is that the authors injected equal volumes (1.5 μ l) of each vector? However, the titers of the stocks varied between 2.4e12 and 1.5e13 according to the labels above the panels, i.e., up to 6.25-fold. If true, the comparison was biased and thus unreasonable, and it should be repeated at the same vector dose (MOI), not volume.

Similarly, in Fig. 2d, were the same volumes (with potentially different titers?) used or the same vector dose?

Also in Fig. 3, the information that "mice were injected with AAV-ie-Atho1 (3.5 E+12 GC/mL)" is not useful and not standard in the field. The crucial information is not the titer of the original stock, but the dose that actually went into the animals.

Shown in Fig. 3c are total numbers, but not percentages, as stated in the legends.

Same in Fig. 4 - please state the final vector dose that was administered (as in Fig. S2), not the titer of the original stock and the volume, which forces readers and reviewers to do their own math.

Same in Fig. S3.

In Fig. S4, panels a and b show the same section, just slightly shifted down in panel b. Shouldn't these be different, since they are from the cochlea versus utricle? Or are the legends wrong?

Besides, also here, the information "were injected 1.5 μ L AAV-ie" (sic!) is not helpful, since the reader needs to know the dose that was administered, not the volume (especially not in the

absence of a titer).

Reviewer #2:

Remarks to the Author:

This short report describes a variant recombinant adeno-associated viral vector AAV-ie (variant of AAV-DJ) that enables efficient gene transfer to supporting cells of the mouse inner ear. This variant includes a cell penetrating peptide associated with the capsid which improves transduction efficiency of supporting cells both in organotypic culture and in vivo, after round window membrane injection at P3.

Efficient viral transduction of supporting cells is an important step towards the development of novel therapies aimed at inducing hair cell regeneration via supporting cell's transdifferentiation or to target mutations in key deafness genes specifically expressed in supporting cells. While the novel vector, here named AAV-ie (for inner ear), is shown to transduce a larger number of supporting cell, substantial work is required to validate this outcome. Shortcomings include: reduced number of animals (averages on n=2 to 3 mice for some experiments, see figure 1- Note that this brings into question the statistical significance of these finding); lack of cross sections that would help identify which supporting cells are transduced and help distinguish hair cells from supporting cells transduction; scanning electron microscopy to validate survival of hair cells and absence of toxicity in vivo; limitation of this study to P17 (P14 for the Atoh1 experiment); etc... The result illustrated in figure 3 is certainly interesting: the authors report improved induction of trans-differentiation of supporting cells into hair cells with AAV-ie-Atoh1 transduction with P0 injections. More extensive analysis of the "regenerated" cells including SEM to evaluate formation of hair bundles would be of interest.

The format of the manuscript is as such that there is no separate introduction or discussion. A more in depth discussion regarding the advantages and limitations of the AAV-ie vector needs to be added. For example, while this vector may be efficient at transducing supporting cells, it also transduce hair cells efficiently (100% IHCs, 60-100% OHCs and 100% utricular hair cells). Use of supporting cell specific promoters or other AAV variants may therefore be necessary to render this approach viable for the treatment of genetic deafness associated with mutations in SC specific genes such as connexins.

Point-to-point responses to reviewer's comments (marked by blue):

The new experimental results are briefed as following:

1. We performed immunofluorescence imaging experiments with equal dose of different serotypes of AAVs, and the corresponding results were included in **Figure.1a** and **Figure. 2a**.
2. We performed immunofluorescence imaging experiments from cross-section preparations, and the corresponding results were included in **Figure. 2d**.
3. We performed a series of SEM to validate the AAV.ie effects on HCs and the SCs overexpression Atoh1, and the new results were included in **Figure. 4b**, **Figure. 4d**, and **Figure. 5e**.
4. We performed electrophysiological experiments to examine the membrane properties of SCs overexpression Atoh1 and other types of cells, and the results were included in **Figure. 5f-i**.

Reviewers' comments:

Reviewer #1 (Remarks to the Author):

In this work, Tan and colleagues have devised various peptide-displaying variants of the previously reported AAV-DJ capsid and then selected one for further validation as a vector for gene transfer to supporting cell in the mouse inner ear.

Overall, I find the data and this work interesting. Unfortunately, however, the way the data are presented makes it impossible for me to truly appreciate the potential value of the AAV-ie vector, for reasons detailed below. Most importantly, I was frequently confused by the changing nomenclature regarding the AAV transduction experiments in which the authors switch back and forth between volumes, doses and titers. In the absence of a unified nomenclature (ideally, vector doses), I cannot fully interpret the shown data and hence cannot scientifically evaluate this work. Fortunately, though, this could be fixed in a revised manuscript provided the editors will make such an invitation. To this end, specific comments and suggestions for improvement are:

Response: We thank the reviewer for the constructive comments. Based on these comments, we have performed additional experiments and analyses and substantially made text and figure revisions, which should have strengthened the quality of our

manuscript.

We agree with the reviewer that it is not ideal that we switch back and forth between volume, doses and titers in the previous submitted manuscript. We performed experiments to use the same dose of different tested AAV variants in order to make our statistical analyses accurate and understandable.

On p.3, the authors mention they inserted five different CPPS, yet Fig. S1a shows only four. Besides, why is the PHP.eB peptide (part of the four) classified as CPP; what are the data supporting this?

Response: We thank the reviewer for pointing out this mistake. We have corrected this part in the revised manuscript. A previous study found that the insertion of this peptide from PHP.eB into capsid can help the vector cross the blood-brain barrier², so we speculated that this peptide could function as a CPP. To accurately reflect the meaning, we have changed the description “CPP” into “CPP-like peptide”.

What was the rationale for pre-screening these variants in HEK293T (kidney) cells and then taking the lead candidates from this screen forward to the studies in the actual target cells? How predictive, if at all, are HEK293T cells for studies in the inner ear? Why not test all vectors directly in the on-target cells? Has this been done, and, if so, what were the results?

Response: We thank the reviewer for raising questions to the design of our experiments. The major purpose of our project is to develop a novel AAV to highly transduce inner ear cells, given that in auditory field there is a high need to develop an ideal AAV variant to effectively transduce the inner ear cells since existing AAVs are not as successful as needed. Here, we took the strategy to insert some peptides which may help AAV to cross membrane. As a first step, we tested the transducing efficiency in HEK293T cells which can give us a clue whether our modification increases the transducing efficiency (**Supplementary Fig. 1**). Among the tested peptides, the CPP-like peptide (DGTLAVPFK in the AAV-ie variant) insertion showed the promise to effectively transduce cells (**Table R1**). After gaining this knowledge, we then directly tested in the on-target cells of the ex-vivo cochlea culture (**Fig. 1**). By this approach, we aim to shorten the time to develop a new AAV variant

to effectively transduce inner ear cells. Indeed, AAV.ie showed a better transducing efficiency to test AAV variants, including Anc80L65 (Fig. 1 and Fig. 2). We added the Table R1 to Supplementary Fig. 1.

AAV	Titer
AAV-DJ	2.8E+12
AAV-ie	2.4E+12
AAV-TAT	7.2E+10
AAV-SynB3	9.2E+9
AAV-PTD-4	1.1E+10

Table R1. Physical particle packaging titers (GC/mL) of AAV variants and AAV-DJ. Average packaging titers from at least two packaging experiments.

At the bottom of page 4, I was confused. In the sentence starting with "However, a large number of ...", are the authors now talking about the vector-treated cohort (since they were referring to the control group in the sentence before)?

Response: We thank the reviewer for pointing out the confusing content in this place. We have changed the sentence to "In the AAV-ie-Atoh1-Nls-mNeonGreen group, a large number of new HCs expressing Myo7a were present in the sensory region as well as in the greater epithelial ridge (GER) region".

In Figure 1, my understanding is that the authors injected equal volumes (1.5 μ l) of each vector? However, the titers of the stocks varied between 2.4e12 and 1.5e13 according to the labels above the panels, i.e., up to 6.25-fold. If true, the comparison was biased and thus unreasonable, and it should be repeated at the same vector dose (MOI), not volume. Similarly, in Fig. 2d, were the same volumes (with potentially different titers?) used or the same vector dose?

Response: We apologize for the confusion here and in the later parts. In **Figure. 1** of previous submitted manuscript, we injected equal volumes of different serotypes of AAVs, but the titers were different. We tried to prove that AAV-ie is more efficient than other AAVs at a lower dose. Recently, AAV-Anc80³ and AAV2.7m8¹ have been identified as powerful vectors in the inner ear. **Both AAVs were compared with other AAVs at different titers, and the titer of AAV-Anc80L65 or AAV2.7m8 was lowest.** Nonetheless, we fully agree with the reviewer that comparing the vectors

with equal doses is more reasonable. Thus, we have performed additional experiments with equal AAV doses to compare the transduction efficiencies among different AAV types. The new results are included in the revised manuscript.

Also in Fig. 3, the information that "mice were injected with AAV-ie-Atho1 (3.5 E+12 GC/mL)" is not useful and not standard in the field. The crucial information is not the titer of the original stock, but the dose that actually went into the animals.

Response: We apologize for the inappropriate description. We have performed additional experiments with higher dose AAV-ie-Atoh1 (1×10^{10} GCs per ear) and added the new data into the revised manuscript (**Fig. 5**). Prompted by the reviewer's constructive information, we have stated the final vector dose instead of titer in the revised manuscript.

Shown in Fig. 3c are total numbers, but not percentages, as stated in the legends.

Response: We thank the reviewer for pointing out this error and have changed to numbers in the revised manuscript.

Same in Fig. 4 - please state the final vector dose that was administered (as in Fig. S2), not the titer of the original stock and the volume, which forces readers and reviewers to do their own math.

Same in Fig. S3.

Response: We thank the reviewer for pointing out this error. We have corrected all places and clearly stated the dose value throughout in the revised manuscript.

In Fig. S4, panels a and b show the same section, just slightly shifted down in panel b. Shouldn't these be different, since they are from the cochlea versus utricle? Or are the legends wrong? Besides, also here, the information "were injected 1.5 μ L AAV-ie" (sic!) is not helpful, since the reader needs to know the dose that was administered, not the volume (especially not in the absence of a titer).

Response: We thank the reviewer for pointing the confusing part in Fig. S4 from the previous submitted manuscript. The images in panel a and b were taken from the same cochlea. Panel a showed AAV-ie infection in **Sox2** positive supporting cells, and panel b showed AAV-ie infection in **Myo7a** positive hair cells. To avoid confusion

and misunderstanding, we have done additional experiments and included the new results in the revised figure. We modified the figure legend to describe this in an accurate and clear way. In addition, we have stated the vector dose in the revised manuscript.

Reviewer #2 (Remarks to the Author):

This short report describes a variant recombinant adeno-associated viral vector AAV-ie (variant of AAV-DJ) that enables efficient gene transfer to supporting cells of the mouse inner ear. This variant includes a cell penetrating peptide associated with the capsid which improves transduction efficiency of supporting cells both in organotypic culture and in vivo, after round window membrane injection at P3. Efficient viral transduction of supporting cells is an important step towards the development of novel therapies aimed at inducing hair cell regeneration via supporting cell's transdifferentiation or to target mutations in key deafness genes specifically expressed in supporting cells. While the novel vector, here named AAV-ie (for inner ear), is shown to transduce a larger number of supporting cell, substantial work is required to validate this outcome.

Response: We are encouraged by the reviewer's comments: "**Efficient viral transduction of supporting cells is an important step towards the development of novel therapies...**". We fully agreed that substantial work is needed to validate such a conclusion. Now we have completed the series of experiments and included the new results to validate our conclusion.

Shortcomings include: reduced number of animals (averages on n=2 to 3 mice for some experiments, see figure 1- Note that this brings into question the statistical significance of these finding);

Response: We thank the reviewer for the comment. Additional experiments were done to increase the number of animals to get the statistical significance of our findings. As **reviewer 1** pointed out that comparing the different AAV vectors with equal doses is more reasonable, we have performed new experiments with equal AAV doses and redid the Figure.1 with new data in the revised manuscript.

lack of cross sections that would help identify which supporting cells are transduced and help distinguish hair cells from supporting cells transduction;

Response: We thank the reviewer for this excellent suggestion. We performed the imaging experiments in the cross sections to investigate the types of supporting cells transduced by AAV.ie, and the new results were added to the revised manuscript (**Fig.2d**). The imaging experiments from cross sections clearly show that AAV.ie transduces all types of SCs.

scanning electron microscopy to validate survival of hair cells and absence of toxicity in vivo; limitation of this study to P17 (P14 for the Atoh1 experiment); etc...

Response: We thank the reviewer for the constructive suggestion of SEM experiments to directly visualize the HC surface. Prompted by this comment, we performed SEM to visualize the surface and morphology of hair cells. We did not find substantial change of HCs, in line with the results of ABR test which showed that AAV.ie did not affect the auditory function. As suggested by the reviewer, we have harvested the cochlea at P30 to evaluate the long-term effects of the vectors after injection the AAV-ie at P3. These new results were included in **Figure. 4** and **Supplementary Fig. 6**.

The result illustrated in figure 3 is certainly interesting: the authors report improved induction of trans-differentiation of supporting cells into hair cells with AAV-ie-Atoh1 transduction with P0 injections. More extensive analysis of the “regenerated” cells including SEM to evaluate formation of hair bundles would be of interest.

Response: We thank the viewer for constructive suggestions of more extensive analysis and SEM experiments. Here, we did three complementary types of experiments to verify the cell identity, immunofluorescence experiments to show the F-actin bundle (**Fig. 5d**), electrophysiological experiments to show the cell membrane properties (**Fig. 5f-i**), and SEM experiments to show the cell morphology (**Fig. 5e**). These observations suggest that SCs converted to HCs by Atoh1 overexpression represent a primitive type of HCs.

The format of the manuscript is as such that there is no separate introduction or

discussion. A more in depth discussion regarding the advantages and limitations of the AAV-ie vector needs to be added. For example, while this vector may be efficient at transducing supporting cells, it also transduce hair cells efficiently (100% IHCs, 60-100% OHCs and 100% utricular hair cells). Use of supporting cell specific promoters or other AAV variants may therefore be necessary to render this approach viable for the treatment of genetic deafness associated with mutations in SC specific genes such as connexins.

Response: We thank the reviewer for the suggestion to modify our manuscript. We have changed the format of the paper and described the advantages and limitations of the AAV-ie vector.

References

1. Isgrig, K. et al. AAV2.7m8 is a powerful viral vector for inner ear gene therapy. *Nature communications* **10** (2019).
2. Chan, K.Y. et al. Engineered AAVs for efficient noninvasive gene delivery to the central and peripheral nervous systems. *Nat Neurosci* **20**, 1172-+ (2017).
3. Landegger, L.D. et al. A synthetic AAV vector enables safe and efficient gene transfer to the mammalian inner ear. *Nature biotechnology* **35**, 280-+ (2017).
4. Yang, J., Bouvron, S., Lv, P., Chi, F. & Yamoah, E.N. Functional features of trans-differentiated hair cells mediated by Atoh1 reveals a primordial mechanism. *J Neurosci* **32**, 3712-3725 (2012).

Reviewers' Comments:

Reviewer #1:

Remarks to the Author:

I thank the authors for addressing most of my comments in a satisfactory way, which has resulted in a significant improvement of their work.

Reviewer #2:

Remarks to the Author:

The revised manuscript is quite different from the initial submission with substantial changes and added experiments performed in a record time frame. With regard to comments from the first round of review, the authors have addressed the major concerns that were raised and they have included new experimental work which was either requested or suggested.

- 1- Number of animals has been increased to n=3
- 2- Added cross section to determine which supporting cells are transduced
- 3- SEM and hair cell count validating lack of toxicity
- 4- SEM of newly generated hair cells

The manuscript, also now includes introduction, results, discussion and methods. In general, it is greatly improved. However, through read through and rewrite of the some sections of the manuscript are necessary. Indeed there are several typos (AAV-ieNeomGreen instead of mNeoGreen and Atoh1 instead of Atoh1... appear in several places, Spinal versus spiral, etc...).

Consistency in the AAV labeling, once with the entire description followed by short name used in the rest of the manuscript. Example: AAV-ie-CAG-Atoh1-Nls-mNeoGreen (AAV-ie-Atoh1). Several sentences are written in poor English, sometimes in a convoluted and confusing way.

Examples: Top of page 5: "SCs are capable of trans-differentiating HCs by manipulating signaling pathways and transcription factors such as gene Atoh1". The way this is written leads to incorrect claims. SCs are not capable of doing anything. Instead, manipulation of signaling pathways... can lead to transdifferentiation of SC into HCs....

Page 6- Conclusion: "AAV-ie represents a big forward step by effectively transducing SCs."

Followed by "Further studies to optimize the specific transducing either SCs or HCs of AAV.ie and to validate these results in large animal models and in animal disease models will be needed for AAV.ie to useful in gene therapy for hereditary auditory diseases." Not proper English here.

Tense is not always consistently used (SEM methods section for example).

Regarding the science, one underestimated result that needs to be expanded is with regard to injections in adult cochlea. Most published work has showed reduced AAV transduction in hair cells for injection in older mice. Supplemental figure 4 shows images from P30 injections and demonstrates high hair cell and supporting cell transduction with AAV-ie. This result should be presented in the main manuscript (not supplemental material) along with cell count of transduction rate for HCs and SCs along the organ (Base, Mid and Apical region). Also, please clarify. If injections were performed at P30, when were the tissues harvested?

Discussion needs improvement. There are no mentions of SCs specific promoters and in general emphasis on the need for SCs specific AAV should be one of the first point to be discussed. The first sentence in the discussion is also very misleading (even though the next sentence cite successful transduction with the Anc80 vector).

Responds to reviewers,

We are pleased to know that our revision with new experimental results answered almost all concerns and comments made by reviewers. We greatly appreciate the specific comments from reviewer 2 which certainly make our manuscript improved further for the publication level of Nature Communications. Following are our response to reviewer's comments.

Reviewer

1. Indeed there are several typos (AAV-ie-NeomGreen instead of mNeoGreen and Atho1 instead of Atoh1... appear in several places, Spinal versus spiral, etc...).

Response: We thank the reviewer for pointing out these typos. We corrected all the wrong spellings in the revision of the paper.

2. Consistency in the AAV labeling, once with the entire description followed by short name used in the rest of the manuscript. Example: AAV-ie-CAG-Atoh1-Nls-mNeoGreen (AAV-ie-Atoh1).

Response: We thank the reviewer for the advice. We have modified the name of AAV-ie-CAG-Atoh1-NLS-mNeoGreen into AAV-ie-Atoh1 as abbreviation.

3. Several sentences are written in poor English, sometimes in a convoluted and confusing way.

Examples: Top of page 5: "SCs are capable of trans-differentiating HCs by manipulating signaling pathways and transcription factors such as gene Atoh1". The way this is written leads to incorrect claims. SCs are not capable of doing anything. Instead, manipulation of signaling pathways... can lead to transdifferentiation of SC into HCs....

Response: We very appreciate the reviewer for the suggestion. We changed the sentence as: Manipulation of signaling pathways and transcription factors such as gene *Atoh1* can lead to transdifferentiation of SC into HCs.

Page 6- Conclusion: "AAV-ie represents a big forward step by effectively transducing SCs." Followed by "Further studies to optimize the specific transducing either SCs or HCs of AAV.ie and to validate these results in large animal models and in animal disease models will be needed for AAV.ie to useful in gene therapy for hereditary auditory diseases." Not proper English here.

Response: We thank you for the comment. We deleted this improper sentence to avoid any confusion.

4. Tense is not always consistently used (SEM methods section for example).

Response: Sorry for the mistake. We revised the SEM methods and checked the tense in other methods sections.

5. Regarding the science, one underestimated result that needs to be expanded is with regard to injections in adult cochlea. Most published work has showed reduced AAV transduction in hair cells for injection in older mice. Supplemental figure 4 shows images from P30 injections and demonstrates high hair cell and supporting cell transduction with AAV-ie. This result should be presented in the main manuscript (not supplemental material) along with cell count of transduction rate for HCs and SCs along the organ (Base, Mid and Apical region). Also, please clarify. If injections were performed at P30, when were the tissues harvested?

Response: We thank the reviewer for these excellent comments. We added the adult data into Figure 2 and clarified the time points of virus injection and tissues harvest in the corresponding figures legends.

6. Discussion needs improvement. There are no mentions of SCs specific promoters and in general emphasis on the need for SCs specific AAV should be one of the first point to be discussed. The first sentence in the discussion is also very misleading (even though the next sentence cite successful transduction with the Anc80 vector).

Response: We thank the reviewer for these comments. We revised the discussion section and added a paragraph to discuss the SCs specific promoters and SCs specific AAV according to the suggestion. To avoid the misleading we deleted the first sentence in the discussion.